# Prevalence of systemic antibacterial use during pregnancy worldwide: A systematic review

Fernando Silva Guimarães[1]*, Tatiane da Silva Dal-Pizzol[2], Marysabel Pinto Telis Silveira[3], Andréa Dâmaso Bertoldi[1]

1 Post-graduate Program in Epidemiology, Federal University of Pelotas, Pelotas, Brazil, 2 Post-graduate Program in Epidemiology, Federal University of Rio Grande do Sul, Porto Alegre, Brazil, 3 Multicenter Post-graduate Program in Physiological Sciences, Federal University of Pelotas, Pelotas, Brazil

☯ These authors contributed equally to this work.

* guimaraes_fs@outlook.com

**Data Availability Statement:** All relevant data are within the manuscript and it Supporting Information files.

**Funding:** The author(s) received no specific funding for this work.

## Abstract

### Objectives

In this study, we aimed to systematically review the literature of the prevalence of systemic antibacterial use during pregnancy and to perform a descriptive analysis focused on methodological characteristics.

### Materials and methods

This study was registered in PROSPERO under protocol number CRD42022376634. Medline, Embase, Scientific Electronic Library Online, *Biblioteca Virtual em Saúde*, Cumulative Index to Nursing and Allied Health Literature, and Web of Science databases were searched (published studies until November 3rd, 2022). Selected studies were population-based cross-sectional or cohort, carried out with pregnant women, and providing information about the prevalence of systemic antibacterial use at least in one trimester of pregnancy. Reviewers conducted in pairs the title and abstract screening, eligibility criteria check, and data extraction of selected studies. Quality appraisal was performed with an adapted version of the Joanna Briggs Institute Critical Appraisal Checklist for Prevalence Studies. Data of included studies were pooled into a graphical and tabular summary.

### Results

A total of 16,251,280 pregnant women and 5,169,959 pregnancy registers were identified. The prevalence estimates of systemic antibacterial use during pregnancy ranged from 2.0% (95%CI 2.0–2.0) to 64.3% (95%CI not reported) in the 79 included studies. The majority were performed in high-income countries (91.5%). Overall, the studies revealed considerable prevalence heterogeneity in terms of study type and dataset used. The 95% confidence intervals were not reported in 41% of studies.

**Competing interests:** The authors have declared that no competing interests exist.

## Conclusion

The disparities in the prevalence of systemic antibacterial use during pregnancy can be related to methodological issues and different health policies. Lack of uniform databases and changes in data collection methods over time should be taken into account in public health strategy planning. The scarce evidence in low- and middle-income settings hampers the comprehensiveness of the global prevalence of antibacterial use during pregnancy.

## Introduction

Antibacterial use during pregnancy is a clinical practice that has been used across different healthcare contexts [1]. It is estimated that approximately 80% of all prescribed drugs in pregnancy include an antibacterial, and up to 25% of women will receive an antibacterial during pregnancy [2]. Respiratory and urinary tract infections (RTIs and UTIs) are the prevailing indications for prescribing these medicines during pregnancy in the outpatient setting [3]. When untreated, these infections are associated with fetal risk outcomes, including neonatal sepsis, spontaneous abortion, premature birth, low birth weight, and chronic lung disease [4]. In fact, as with the use of any drug during pregnancy, prescribing antibacterials is a risk-versus-benefit decision, where it may challenge the physician's choice due to the absence of safety and efficacy data, usually available from randomized controlled trials, considered not feasible and unethical in pregnant women [2].

Despite these concerns, the inappropriate use of antibacterials can lead to antimicrobial resistance (AMR), which has been postulated among the ten greatest threats to global health, by the World Health Organization (WHO) [5]. AMR continues to be a major public health issue in middle and high-income countries, whereas the consumption of antibacterials, considering the general population, increased by 110% from 2000 until 2015 [6]. AMR is also associated with 30% of deaths from neonatal sepsis worldwide [7] and is primarily driven by antibacterial misuse in low- and middle-income countries (LMICs) [6]. The Global Action Plan on Antibacterial Resistance stated by WHO [8] recommends that all countries must collect and report antibacterial consumption data in the general population. However, there is uncertainty about patterns of antibacterial use regarding special populations, such as pregnant women, and this factor can impact the average national consumption, primarily in LMICS.

To date, reviews of antibacterial use during pregnancy were performed along with the overall drug use during pregnancy, and have focused only on developed countries [9,10]. This is a particularly important gap as the LMICs may influence the antibacterial consumption during pregnancy, considering AMR and different patterns of antenatal drug use in these countries, which depends on clinical practices guidelines, health systems frame, among other factors [11]. The evidence for prescribing antibacterials in pregnancy is limited in LMICs, compared to developed countries, which are mostly based on electronic prescription databases [8]. Despite this difference, estimates from LMICs can provide an overview of the likely prevalence of antibacterial use during pregnancy, addressing the needs of the AMR action plans in LMICs [6,8]. Further information on trends of antibacterial use and indication is also needed.

Furthermore, performing a systematic review focused on the proportion of a population currently affected with a particular status of interest–use of antibacterial during pregnancy—can contribute to informing healthcare professionals, as well as policymakers to plan and manage the burden of antibacterial misuse during pregnancy [12]. A tabular summary of prevalence point estimates can provide useful information of evidence synthesis [12]. Thus,

gathering data on the proportion of antibacterial use during pregnancy enables the comparability between subgroups of studies, in order to identify point estimates variability according to relevant characteristics, such as study type, data source and country income. Additionally, the WHO global action plan on AMR emphasizes the role of academic research to strengthen the evidence base of AMR and antibacterial use through information on prevalence and geographical patterns [8].

To evaluate the patterns of antibacterial use during pregnancy, we undertook a systematic review of studies that had information about the prevalence of systemic antibacterial use during pregnancy. Specifically, we aimed to (1) describe the prevalence of systemic antibacterial use during pregnancy and (2) perform a descriptive analysis to address differences between study-level characteristics.

## Materials and methods

This study was reported according to the Preferred Reporting Items for Systematic Review and Meta-Analyzes (PRISMA) [13] and registered in the International Prospective Register of Systematic Reviews (PROSPERO) under the protocol number CRD42022376634 [14]. The Cochrane Handbook for Systematic Reviews of Interventions [15] was used throughout the review and adjusted for systematic review of prevalence.

### Eligibility criteria

Selected studies were population-based cross-sectional or cohort original studies, carried out with pregnant women living in any country–regardless of the income level–and included at least information about the prevalence of systemic antibacterial use in the first, second, or third trimester of pregnancy. There were no constraints on follow-up time, date, or language.

Studies were deemed ineligible when the data related to antibacterial use during pregnancy was limited to a single type of antibacterial or pharmacological class, or in situations where it was not possible to ascertain if the antibacterials were systemic. Also, studies with non-representative samples, performed with animals, case-control studies, case reports, randomized clinical trials, reviews and systematic reviews, commentaries, letters, abstracts, and qualitative studies were excluded.

### Information sources and search strategy

Six electronic databases were searched, namely: Medline (PubMed), Embase, Scientific Electronic Library Online (SCIELO), *Biblioteca Virtual em Saúde* (BVS), Cumulative Index to Nursing and Allied Health Literature (CINAHL), and Web of Science, from inception to November 2022. Search strategies included terms related to antibacterial use, pregnancy, and study design. Table 1 display the primary search strategy structured in Medline (PubMed), which supported the remaining databases search strategies, as shown in S1 Table. Reference lists of included studies were screened to identify potentially eligible studies that were not located and identified in the databases. Additionally, we scanned the references of reviews and systematic reviews given by the search strategy. We contacted authors of non-located papers using e-mail and social media sites, such as LinkedIn, Academia.edu, and ResearchGate.

### Selection and data collection processes

Three reviewers (FSG, TSD, and MPS) screened articles in pairs by title and abstract, using a free web tool (Rayyan) [16] to assist in screening and selecting studies for systematic reviews. In the next stage, the articles were checked by full-text to determine the final inclusion decision

**Table 1. Medline/PubMed search strategy to identify antibacterial use during pregnancy studies.**

| # | Search strategy |
|---|---|
| 1 | ("anti bacterial agents/therapeutic use"[MeSH Major Topic] OR "anti bacterial agents/administration and dosage"[MeSH Major Topic] OR "Anti-Bacterial"[Title/Abstract] OR "Anti-Bacterial"[Title/Abstract] OR "Antibacterial"[Title/Abstract] OR "bacteriocid*"[Title/Abstract] OR "antibiotic*"[Title/Abstract] OR "Antimicrobial"[Title/Abstract] OR "drug prescriptions"[MeSH Terms] OR "drug utilization"[MeSH Terms] OR "prescrib*"[Title/Abstract] OR "prescription*"[Title/Abstract] OR "drug utilization"[Title/Abstract] OR "drug utilisation"[Title/Abstract] OR "drug use*"[Title/Abstract]) |
| 2 | ("pregnancy"[MeSH Major Topic:noexp] OR "pregnancy trimesters"[MeSH Terms] OR "pregnant women"[MeSH Terms] OR "prenatal care"[MeSH Terms] OR "pregnan*"[Title/Abstract] OR "Prenatal"[Title/Abstract] OR "Antenatal"[Title/Abstract] OR "gestation*"[Title/Abstract]) |
| 3 | ("Prevalence"[MeSH Terms] OR "Incidence"[MeSH Terms] OR "surveys and questionnaires"[MeSH Terms:noexp] OR "Health Surveys"[MeSH Terms:noexp] OR "Epidemiologic Studies"[MeSH Terms:noexp] OR "Cohort Studies"[MeSH Terms] OR "Cross-Sectional Studies"[MeSH Terms] OR "epidemiology"[MeSH Subheading] OR "epidemiolog*"[Title/Abstract] OR "observational"[Title/Abstract] OR "prevalen*"[Title/Abstract] OR "Incidence"[Title/Abstract] OR "survey*"[Title/Abstract] OR "questionnaire*"[Title/Abstract] OR "cohort"[Title/Abstract] OR "frequency"[Title/Abstract] OR "follow-up"[Title/Abstract] OR "followup"[Title/Abstract] OR "longitudinal"[Title/Abstract] OR "prospective"[Title/Abstract] OR "retrospective"[Title/Abstract] OR "cross-sectional"[Title/Abstract] OR "population-based"[Title/Abstract]) |
| 4 | 1 and 2 and 3 |

regarding the eligibility criteria. At each screening stage, any differences between the two reviewers were discussed, and a fourth reviewer (ADB) was consulted for a consensus decision for eligibility and inclusion. In addition, FSG, TSD, and MPS executed data extraction independently from the final list of selected articles using an extraction table.

## Data items

The extraction table included information on studies regarding author and year, study type (cross-sectional/ cohort), location (country), country income (high/ low and middle), dataset (primary/ secondary/ primary and secondary), antibacterial classification system used (Anatomical Therapeutic Chemical (ATC) [17]/ others), type of denominator (pregnant women/ pregnancies/ mother-child dyad), maternal schooling (years of schooling), mean age at birth (years), cesarean (yes/ no), focused on prevalence (studies with prevalence as the main objective/ other objectives such as association or causal effect, and comparative methodology studies). The sample size, and the number of women exposed to systemic antibacterial during pregnancy in the whole pregnancy period, and at 1st, 2nd, or 3rd trimesters were identified for each study (S2 Table). Information on the proportion of antibacterial subgroups of studies included is described in S3 Table. The number and proportion of the following subgroups were summarized: beta-lactam; sulphonamides and trimethoprim; macrolides, lincosamides and streptogramins; tetracyclines; quinolones; nitrofuran derivatives and imidazole. The supplementary material of each included study was checked for instances of unclear or missing information. Data extraction was performed by FSG and double-checked by the reviewers.

## Risk of bias assessment

Quality appraisal was conducted using an adapted version of the Joanna Briggs Institute Critical Appraisal Checklist for Prevalence Studies (JBI) [12]. Based on the literature [18,19], the adapted version consisted of the following additional information: 1) Check supplementary materials for pregnant women data in studies with mother-child dyad sample; 2) Use of

methods to deal with a complex survey design (i.e., survey weights); 3) Sample-size assessment was not required in nationally secondary data studies; 4) Study sample described in descriptive table or text; 5) Report response rates (i.e., proportion of pregnant women who answered the survey divided by the number of eligible pregnant women) in a flowchart or text; 6) Validated instruments (i.e., training and comparative data of interviewers, quality control data or pilot study) were considered for cohort or cross-sectional studies. Consistency in data collection, report of missing data or methods to estimate data validation were considered for secondary data studies; 7) Properly identification of systemic antibacterials (i.e., stated by authors or ATC J01); 8) Association studies without confidence intervals were appraised as "not applicable"; 9) Non-response sociodemographic description or non-differential losses analysis. An adequate response rate was defined with a cut-off point $\geq$ 80%.

FSG conducted the quality appraisal, and a sample of studies was double-checked by the reviewers, to evaluate the judgment criteria. The tool contained 9 questions regarding the study design, with the following answer options: "Yes" indicating higher quality, "No" indicating poor quality, "Unclear" indicating absence of information and "Not applicable" indicating the studies' unfit for criteria evaluation. The questions comprehended the appropriate sample frame and sampling process, adequate sample size, participants and context, appropriate coverage in data analysis, valid methodology, condition of outcome measurement and information on response rate (S4 Table). No exclusions were made based on the overall quality of the studies.

### Data synthesis

Data were pooled into a tabular summary according to author and year of publication, study type, country, dataset, type of denominator, maternal schooling, maternal age at birth, cesarean, denominator (N) and number of women exposed to antibacterial during pregnancy (n). The subgroup descriptive analysis according to type of study, dataset, type of denominator, sample size, studies focused on prevalence (not association, causal effect, or comparative methodology studies), and country income were conducted to describe differences in the proportion range of systemic antibacterial use during pregnancy, considering the overall sample and the $90^{th}$ percentile. The κ statistic was used to evaluate the agreement between reviewers.

A meta-analysis was not feasible due to the high heterogeneity between studies, and there are no specific tests to evaluate this issue in a prevalence meta-analysis [20]. Regarding graphical representations, data of prevalence estimates were gathered according to sample size in the overall sample. Additionally, estimates considering the subgroup analysis for country income and type of dataset were showed, stratified by sample size. Confidence intervals (CI) were presented when reported by the authors of included studies. We described proportions of antibacterial use during pregnancy before and after the Global Action Plan on Antibacterial Resistance stated by the WHO. We identified the most common indications for antibacterial use during pregnancy among the included studies. The statistical analysis was performed using STATA 14.2 (StataCorp., College Station, TX, USA).

### Results

After excluding 11,500 duplicates using the automated tool Rayyan, 18,418 titles and abstracts were screened. We excluded 18,247 records for reasons such as study type according to exclusion criteria and theme out of scope, resulting in 164 records assessed for eligibility, leading to 77 included studies. All included studies were published in English language. The exclusion criteria were related to sampling processes (16), lack of antibacterial information (65), and study design (6). We screened the references of included studies, resulting in 16 records to

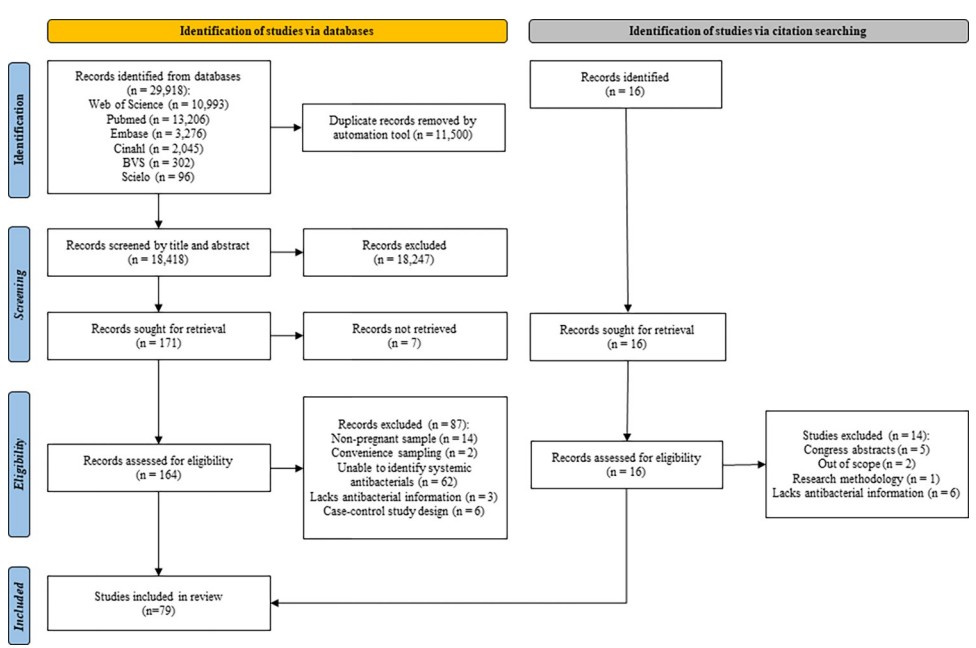

**Fig 1. Preferred Report Items for Systematic Reviews and Meta-Analysis (PRISMA) flow diagram of included studies.**

assess eligibility, including two more studies, reaching a total of 79 studies [3,11,21–97] included in the systematic review (Fig 1). Concerning the evaluation of study eligibility, there was a substantial agreement between reviewers (κ statistic = 0.85).

The pooled characteristics of included studies are shown in S2 Table. A total of 16,251,280 pregnant women and 5,169,959 pregnancy registers were identified in the included studies that were feasible to estimate the prevalence of systemic antibacterial use during pregnancy. Regarding the 79 included studies, 43 provided information on maternal age [3,21–25,27–30,33–35,39,43,44,46,48,50,54,57,58,60,62,65,67,69,71,73–84,88,91,97], 18 presented the proportion of cesarean [3,11,21,23,25,27,39,44,57,59,61,62,63,71,74,87,93,95], 15 articles showed data on maternal schooling [25,27,34–36,39,46,54,61,66,82,87,88,91,96], and 10 for gestational age [21,23,34,35,36,44,51,76,78,93]. Data from the years 2000 and later were used for 57% of the included articles. The majority (n = 73) were performed in high-income countries (91.5%); only 6 were from lower- and upper-middle-income countries (8.5%) [3,36,47,56,87,91]. Amidst high-income studies, Denmark (17%) [21,34,37,39,42,44,45,54,57,64,73,80,42,95] and United States (12%) [27,29,32,40,60,70,75–77] were the two most prevalent countries. Proportions of antibacterial subgroups were shown in S3 Table. The risk of bias assessment is shown in S4 Table.

Concerning quality appraisal, this study reported high rates of quality evidence for sampling characteristics evaluated in questions 1 to 3 (Fig 2). Lower rates of higher quality were also observed for a detailed description of the study sample (76%), standard and reliable measures for antibacterial use during pregnancy (76%), and the use of valid methods (72%). Only 44% of included studies for appropriate coverage in data analysis and 27% for information on response rate were classified as "yes". Confidence intervals were unclear for 41% of the included studies and 52% of studies were "not applicable".

The prevalence of systemic antibacterial use during pregnancy for the 79 included studies ranged from 2.0% (95%CI 2.0–2.0) to 64.3% (95%CI not reported), and the average estimate

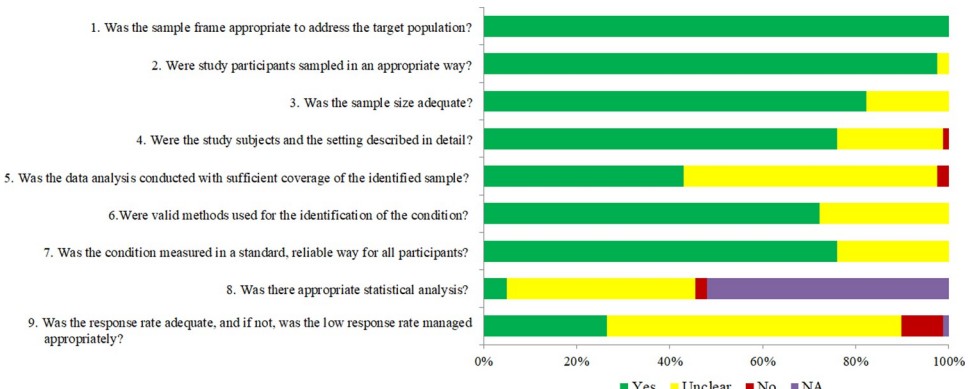

**Fig 2. Frequency of quality assessment categories of the Joanna Briggs Institute critical appraisal tools (adapted version).** NA: Not applicable; U: Unclear.

(27%) is indicated by the line in Fig 3. Table 2 shows the subgroup analysis in the overall sample and 90th percentile according to methodological decisions: type of study, dataset, type of denominator, studies focused on prevalence, and country income. Fig 4 shows the graphical representation for the estimates of antibacterial use during pregnancy for country income and dataset subgroups, according to sample size categories. Subsequently to the Global Action Plan on Antibacterial Resistance stated by WHO in 2016 [8], only three studies [28,52,85] were performed using data collection since 2016, and the proportions ranged from 2.6% (95%CI not reported) to 8.2% (95%CI not reported). Studies with data collected before 2016 showed a prevalence ranging from 2.0% (95%CI 2.0–2.0) to 64.3% (95%CI not reported). Only six studies [11,23,25,28,47,81] provided information on antibacterial indications. Among those, the most common indications were urinary tract infections and respiratory tract infections.

## Discussion

In this systematic review, we identified, summarized, and compared 79 published studies across 29 countries, with a prevalence range in the overall sample from 2.0% (95%CI 2.0–2.0) [47] to 64.3% (95%CI not reported) [29]. With regards to the 90th percentile sample, frequencies varied from 7.0% (95%CI not reported) [45] to 48.3% (95%CI not reported) [87]. Point estimates of antibacterial use during pregnancy amongst LMICs included studies varied from 2.0% (95% CI 2.0–2.0) [47] to 48.3% (95%CI not reported) [87]. A systematic review conducted in outpatient primary care in LMICs showed a range from 19.6% (95% CI 14.0–26.4) to 90.8% (95% CI 89.3–92.0) [98]. Presently, there is a need to generate accurate nationally representative prescribing data of antibacterial use [98], mainly from lower-income countries and at the point of care [8]. We identified only three included studies with data on antibacterial use by trimesters (S2 Table). Unfortunately, we were unable to perform a comprehensive analysis due to the small number of studies with this information.

We employed the flowchart criteria to properly evaluate response rates and exclusions (Fig 2, question 5). However, less than half of the studies reported this information. Moreover, we observed a lower rate of studies that discussed the response rate and reasons for non-response, even though we used a >80% cut-off point as an adequate response rate and discriminatory criteria for primary and secondary data studies [19,20]. We observed substantial uncertainty in identifying confidence intervals (41%), despite the fact that 71% consisted of prevalence studies, for which the 95% CI display would be expected [12,15]. Also, 52% posed

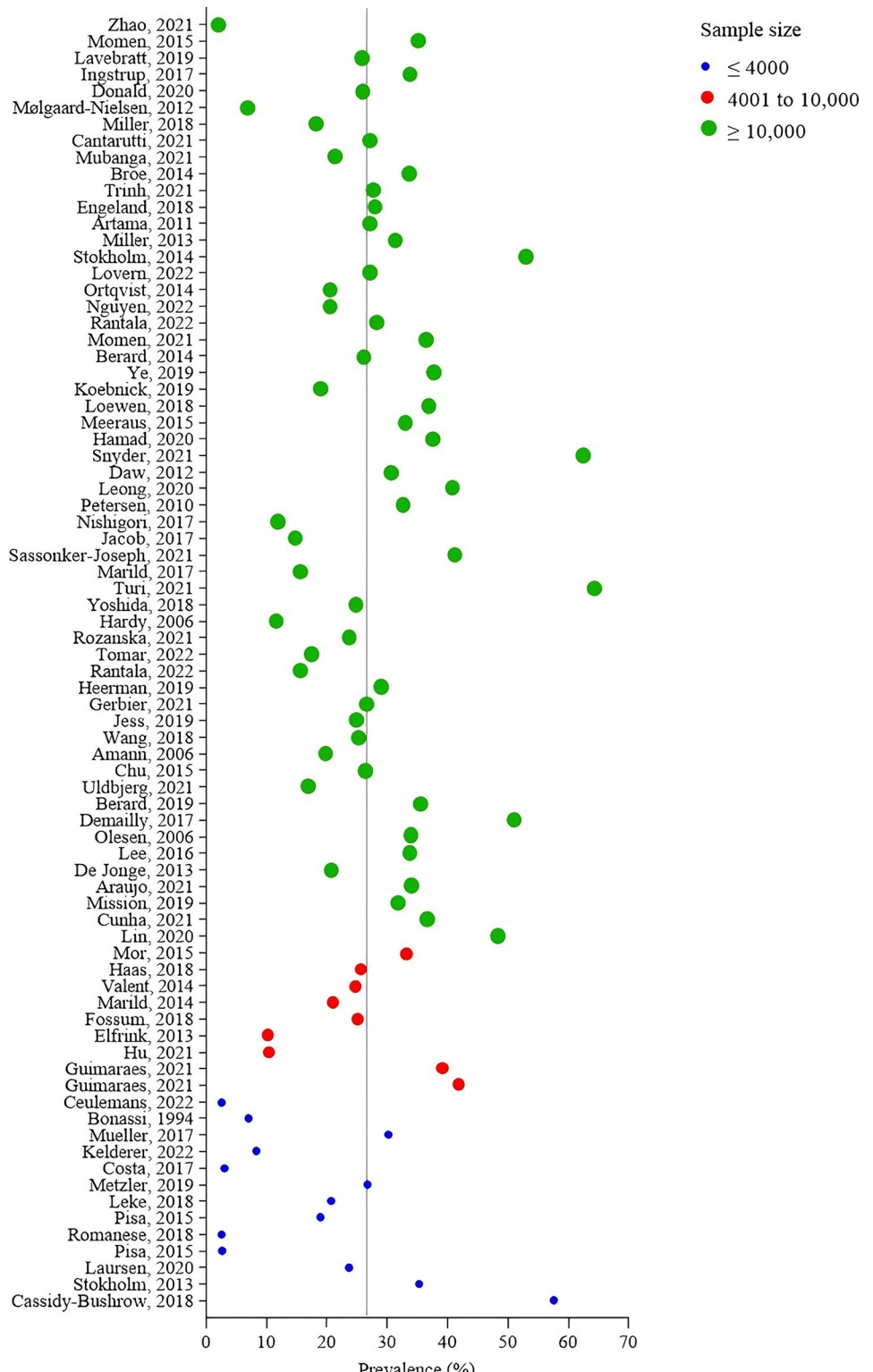

**Fig 3. Estimates of systemic antibacterial use during pregnancy of included studies (n = 79) according to sample size.**

**Table 2. Prevalence range estimates of systemic antibacterial use during pregnancy in the overall sample and 90th percentile according to methodological decisions (study type, type of denominator and studies focused on prevalence) of included studies (n = 79).**

| | N of studies[a] | Prevalence range (%) | |
|---|---|---|---|
| | | Overall sample | p90 |
| **Study type** | | | |
| Cross-sectional | 19 | 2.0–41.8 | 7.0–33.0 |
| Cohort | 60 | 3.0–64.3 | 10.1–50.9 |
| **Dataset[b]** | | | |
| Primary data | 18 | 2.6–41.8 | 7.0–26.6 |
| Secondary data | 60 | 2.0–64.3 | 14.7–48.3 |
| Primary and secondary data | 4 | 2.6–57.5 | 16.9–36.7 |
| **Type of Denominator** | | | |
| Pregnant women | 32 | 2.0–64.3 | 7.0–35.2 |
| Pregnancies | 16 | 10.1–50.9 | 11.5–34.0 |
| Mother-child dyad | 31 | 6.8–64.3 | 15.6–48.3 |
| **Studies focused on prevalence[c]** | | | |
| Yes | 31 | 2.6–62.4 | 7.0–39.1 |
| No | 48 | 2.0–64.3 | 8.2–40.7 |
| **Country income[b]** | | | |
| High | 75 | 2.6–64.3 | 8.2–41.2 |
| Low and middle | 7 | 2.0–48.3 | 20.7–39.2 |

[a] Number of studies regarding the overall sample (n = 79). [b] The number of studies exceeded 79 due to three studies that presented multiple types of data and were accounted for more than once in each category. [c] Association studies, which constituted the majority, and comparative studies, were included in the "No" category. 95% CI: 95% confidence intervals. P90: 90th percentile.

for association studies, being classified as "not applicable" in this question, due to the nature of JBI to evaluate exclusively prevalence studies [12].

The upper limit of estimates in the overall sample and 90th percentile for secondary data studies was higher (64.3% and 48.3%) compared to primary data studies (41.8% and 26.6%). Primary data (i.e., collected through interviews or self-administered questionnaires) studies can provide more accurate measures, given the researcher's role in the data gathered [99,100]. Secondary data studies are prone to inconsistencies in data capture during the study period and can be affected by local drug policies [100]. The Global Action Plan on Antimicrobial Resistance (2015) [8] and Antenatal Care for a Positive Pregnancy Experience (2016) [101], both provided by WHO, are widely comprehensiveness strategies for rational antibacterial use in healthcare settings and the use of antibacterial during antenatal care, respectively.

There is a chance of underestimation regarding primary data study estimates compared to secondary dataset studies, and these need to be properly interpreted [102]. The latter is recorded prospectively and independently, which can help to minimize maternal recall bias [100]. Additionally, well designed secondary data studies depends on the quality of the data [18], while primary studies rely on the recall period and the research question [100]. However, primary and secondary data studies has to deal with misclassification bias, since the information on prescription and dispensing may not reflect the actual consumption [99]. Depending on the type of data, measurement errors can result from different sources: 1) The absence of data for prescription database studies; and 2) The lack of reporting medicine use collected from interviews or self-administered questionnaires in primary data studies [99]. Therefore,

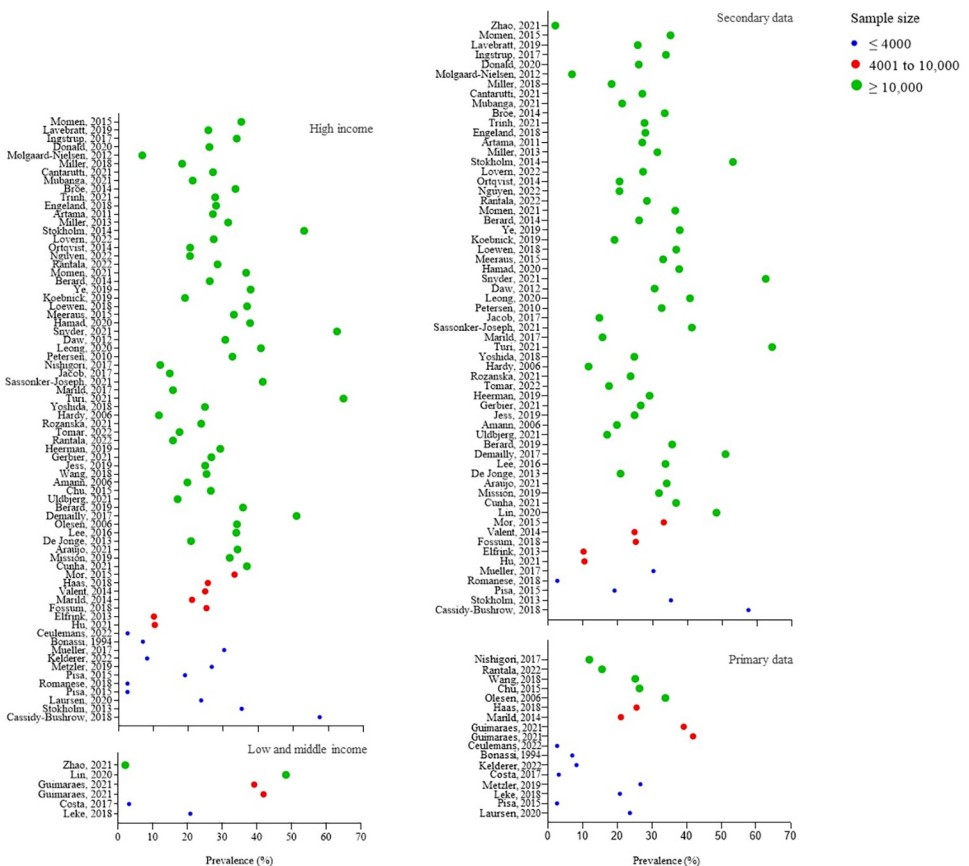

**Fig 4. Estimates of antibacterial use during pregnancy from included studies for country income and dataset subgroups, stratified by sample size.**

both errors could underestimate the prevalence of antibacterial use, although secondary studies with high-quality data may provide more reliable prevalence estimates compared to primary studies [102]. Importantly, the unregistered medicine use in secondary datasets can be assessed from primary data studies, when both types of data are available [100].

Furthermore, another methodological challenge is identifying the exact timing of the pregnancy period, which can be measured through reliable information on the first day of the last menstrual period (LMP), and other estimative methods likewise [102]. The majority of secondary studies included in this systematic review were performed in Denmark [21,34,37,39,42,44,45,54,57,64,73,80,82,96], which estimates the LMP calculated from gestational age data. This method was also used in studies from France [51,77,93] and Norway [3,26,38,41,58,85]. However, the amount of LMP information varies, and estimates are produced using algorithms [102], which increases uncertainty. All studies reported the overall prevalence of antibacterial use throughout the pregnancy period (1st, 2nd, and 3rd trimesters). In general, information on induced terminations and spontaneous pregnancy losses is available in secondary databases [102]. This data can be considered too sensitive to be publicly released for research [102] and may lead to differences in exposure to antibacterials in pregnancy. Additionally, the pregnancy period misclassification resulting from LMP inaccuracy is greater for short-term use characteristic of antibacterials, compared to medicines used to treat chronic conditions, which can probably underestimate the prevalence of antibacterial use during pregnancy [102]. Therefore, the prevalence range divergence between primary and

secondary data studies could be explained, in part, by these methodological differences that could contribute to the high heterogeneity considering included studies.

Given the type of denominator, there was no significant subgroup difference in the prevalence range of antibacterial use during pregnancy. It was expected that the prevalence range of studies with pregnant women would be similar to those with mother-child dyads since they count the total number of pregnant women sampled as the denominator. The lower limit prevalence estimates in the overall sample and 90th percentile was higher for mother-child dyad studies (6.8% and 15.6%), compared to studies with pregnant women (2.0% and 7.0%). Unlike the mother-child dyad samples, studies regarding pregnancy episodes could capture live and stillborn deliveries, induced terminations of pregnancy, and spontaneous pregnancy losses [102]. However, these studies typically require the use of some personally identifiable information and multiple databases, therefore, the data on pregnancy losses could be considered too sensitive to use in research [99].

Our findings revealed significant variations in the prevalence of antibacterial use during pregnancy within countries, reflecting the heterogeneity between studies. In the overall sample, the lowest prevalence of 2.0% (95%CI 2.0–2.0) was reported from a Chinese nationwide cross-sectional study [47], and data of medical records were gathered from 2014 to 2018. Importantly, a substantially lower prevalence does not imply the rational use of antibacterials, since three-quarters of the analyzed antibacterials may have been inappropriate prescribed according to authors. In a similar context, an included study performed in Taiwan [53] reported a prevalence of 33.67% (95% CI not reported) using data of a National Health Insurance database, from 2007 to 2010. The ease of access to antibacterials without prescriptions in the general population remains in China, mainly in retail pharmacies [103]. This has led to a comprehensive action plan of prescription-only antibacterials at pharmacies in all Chinese provinces by 2020 [104]. Therefore, disparities in policies, clinical practices and data sources could result in cross-site variability, besides methodological issues [18].

The highest prevalence of 64.3% (95%CI not reported) in the overall sample was reported from an American study of mother-child dyads with data from the insurance database Tennessee Medicaid Program (TennCare) [29]. We identified studies using insurance, reimbursement, and sales databases. Importantly, the use of sales databases, compared to insurance databases, raises multiple challenges: First, there is a chance of under-detection bias by antibacterial over-the-counter (OTC) sales, likely found in the data collection period (1995 to 2003) where control policies regarding AMR were scarce [6]. Second, chance of data collection bias since the sales database may not cover a large segment of the population not covered, given the lack of data from public sector [105]. Third, the risk of misclassification since the antibacterial use is defined based on filled prescriptions and not drug consumption precisely [106].

Regarding to pregnant women, perceptions and beliefs about the antibacterial use are also relevant. An American qualitative study showed a rise of awareness about antibacterials usage and AMR [107]. Pregnant women widely expressed a concern to avoid taking antibacterials, and a demand to improve health education strategies of antibacterial use during pregnancy (e.g., antibacterial counseling by healthcare providers). The United States had a comprehensive national effort to control AMR, along with Norway and United Kingdom [108]. In addition to cultural norms, the organization of healthcare systems and the decision-making process of antibacterial prescription for pregnant women across countries should be emphasized [109]. Therefore, the high level of heterogeneity in our study can be a result of: 1) the inherently change of point prevalence estimates with regard to time of data collection, location, and evaluated subgroup [20,110]; 2) more variability of point estimates among different studies than for comparative measures such as odds ratio or relative risk [110]; and 3) The variance

estimator and the type of outcome being pooled [20,110]. We included studies without restrictions on follow-up length and period, which could contribute to this level of heterogeneity, as well as the effect of local and national policies of antibacterial use on data capture in secondary databases [100].

We included representative-sample studies, and no restriction regarding follow-up period, date, or language was applied, in order to perform a comprehensive analysis of published studies. Also, the study protocol was published *a priori* to prevent biased reporting. Given the burden of infectious diseases in pregnant women [2–5], antibacterial use during pregnancy accounts for an important part of the general population prevalence, and the results of this study can contribute from a public health perspective.

Our study had some limitations. First, there is a chance of location bias considering the seven records (4%) not retrieved in the screening process (Fig 1). This reporting bias occurs in journals with different ease of access or levels of indexing in standard databases [15]. However, the majority of these studies (n = 6) were not available on database sites, and five of them were published before 2001. The remaining studies were published in 2006 and 2018. Second, we identified 62 studies with information about antibacterial, however, we couldn't classify them as systemic antibacterial. Third, we did not evaluate publication bias through statistical tools. Regarding a prevalence meta-analysis, the use of tools such as funnel plot and Egger's test is not recommended, due to a lack of consensus regarding the definition of a positive and negative result in a meta-analysis of proportions [20]. Fourth, we did not include search terms related to case-control studies. We opted for population-based cohort and cross-sectional studies to summarize the prevalence, compared to nested case-control studies, due to the selection bias that may ensue in case-control designs. Moreover, cross-national comparisons (CNC) studies of antibacterial use during pregnancy are prone to methodological limitations commonly found in drug utilization studies, as the influence of demographic difference between countries in the consumption patterns, lack of uniform databases across different countries and changes in data collection methods over time [105]. Finally, the health system background of each country must be accounted [111], mainly in countries where the provision of healthcare is fragmented [105].

## Conclusion

This systematic review showed a prevalence range of systemic antibacterial use during pregnancy from 2.0% (95%CI 2.0–2.0) to 64.3% (95%CI not reported). Overall, the studies revealed considerable heterogeneity in terms of methodological characteristics. The 95% confidence intervals were not reported in 41% of studies. The scarce evidence in low- and middle-income settings hampers the comprehensiveness of the global prevalence of antibacterial use during pregnancy. These findings should be considered in planning public health strategies and encourage data generation regarding this theme in low- and middle-income countries. Efforts to confront structural inequalities in health research and to promote equitable data sharing are required in order to support evidence of antibacterial use in such contexts [104,105].

## Supporting information

**S1 File. PRISMA 2020 checklist.**
(PDF)

**S1 Table. Search strategy to identify systemic antibacterial use during pregnancy studies.**
(PDF)

**S2 Table. Characteristics of studies included in the systematic review.**
(PDF)

**S3 Table. Proportion of antibacterial subgroups of studies included in the systematic review.**
(PDF)

**S4 Table. Quality assessment of included studies (n = 79) using the Joanna Briggs Institute critical appraisal tools (adapted version).**
(PDF)

**S5 Table. Data extraction summary and eligibility confirmation of included studies (n = 79).**
(PDF)

## Acknowledgments

We are thankful to all participants and researchers of the studies included in this systematic review.

## Author Contributions

**Conceptualization:** Fernando Silva Guimarães, Tatiane da Silva Dal-Pizzol, Marysabel Pinto Telis Silveira, Andréa Dâmaso Bertoldi.

**Data curation:** Fernando Silva Guimarães, Tatiane da Silva Dal-Pizzol, Marysabel Pinto Telis Silveira, Andréa Dâmaso Bertoldi.

**Formal analysis:** Fernando Silva Guimarães.

**Investigation:** Fernando Silva Guimarães, Tatiane da Silva Dal-Pizzol, Marysabel Pinto Telis Silveira, Andréa Dâmaso Bertoldi.

**Methodology:** Fernando Silva Guimarães, Tatiane da Silva Dal-Pizzol, Marysabel Pinto Telis Silveira, Andréa Dâmaso Bertoldi.

**Project administration:** Fernando Silva Guimarães.

**Supervision:** Fernando Silva Guimarães, Tatiane da Silva Dal-Pizzol, Andréa Dâmaso Bertoldi.

**Validation:** Fernando Silva Guimarães, Tatiane da Silva Dal-Pizzol, Marysabel Pinto Telis Silveira, Andréa Dâmaso Bertoldi.

**Visualization:** Fernando Silva Guimarães, Tatiane da Silva Dal-Pizzol, Marysabel Pinto Telis Silveira, Andréa Dâmaso Bertoldi.

**Writing – original draft:** Fernando Silva Guimarães, Tatiane da Silva Dal-Pizzol, Marysabel Pinto Telis Silveira, Andréa Dâmaso Bertoldi.

**Writing – review & editing:** Fernando Silva Guimarães, Tatiane da Silva Dal-Pizzol, Marysabel Pinto Telis Silveira, Andréa Dâmaso Bertoldi.

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
