## [Decision Letter · Decision Letter 0]

5 Jan 2024

PONE-D-23-25313Prevalence of systemic antibacterial use during pregnancy worldwide: a systematic reviewPLOS ONE

Dear Dr. Guimarães,

Thank you for submitting your manuscript to PLOS ONE. After careful consideration, we feel that it has merit but does not fully meet PLOS ONE’s publication criteria as it currently stands. Therefore, we invite you to submit a revised version of the manuscript that addresses the points raised during the review process.

We look forward to receiving your revised manuscript.

Kind regards,

Nur Aizati Athirah Daud, Ph.D.

Academic Editor

PLOS ONE

Journal Requirements:

2. We note that your Data Availability Statement is currently as follows: “All relevant data are within the manuscript and it Supporting Information files”

Reviewers' comments:

Reviewer's Responses to Questions

**Comments to the Author**

1. Is the manuscript technically sound, and do the data support the conclusions?

Reviewer #1: Yes

Reviewer #2: Partly

2. Has the statistical analysis been performed appropriately and rigorously? 

Reviewer #1: N/A

Reviewer #2: N/A

3. Have the authors made all data underlying the findings in their manuscript fully available?

Reviewer #1: No

Reviewer #2: Yes

4. Is the manuscript presented in an intelligible fashion and written in standard English?

Reviewer #1: Yes

Reviewer #2: No

5. Review Comments to the Author

Reviewer #1: The review highlighted a wide range of antibiotic usage during pregnancy, with prevalence rates ranging from as low as 2% to as high as 64.3%. Notably, in low- and middle-income countries (LMICs), antibiotic use during pregnancy was observed to be between 2% and 48.3%, while outpatient primary care settings in LMICs showed prevalence rates spanning from 19.6% to 90.8%. Additionally, the review shed light on the challenges encountered in conducting this analysis, particularly in terms of limitations associated with the available data and the lack of comprehensive information provided by the included studies.

In my perspective, this review offers valuable insights, particularly regarding the epidemiology of antibiotic use during pregnancies.

Nevertheless, there are several areas where additional information could enhance our understanding. Firstly, it would be beneficial to examine the prevalence of antibiotic use both before and after the implementation of the WHO's AMR global plan. Additionally, investigating the common indications for antibacterial usage and stratifying prevalence based on these indications could provide valuable context.

Moreover, among the 79 included studies, it would be informative to categorize the types of antibiotics used, potentially utilizing the WHO's classification and categories for a more comprehensive analysis. Furthermore, Table 2 presents an opportunity to explore the prevalence of antibiotic use during different trimesters of pregnancy, allowing for a nuanced stratification and summary of this data. Addressing these aspects could contribute to a more comprehensive understanding of antibiotic usage patterns during pregnancy.

Other comments

Methodology:

• How about studies reporting in different languages? How did the authors process or analyse the article? Any translation done?

• How did the CI obtain? Was it from the value reported in the included studies or any additional analysis done?

• In discussion, what does authors mean by response rate?

Reviewer #2: The authors present an impressive exercise that attempts to generate estimates of for the prevalence of systemic antibacterial use during pregnancy. As a reviewer, I recognize the public health impact of this question, given different global initiatives to address antibiotic resistance. The manuscript is well organized, and the authors’ objective is clear. The manuscript could benefit from clarification in several sections. Specifically, the authors discuss different biases in the use of primary vs secondary data, 1) without providing information how these biases would affect prevalence estimates and 2) that may not be relevant in the context of these analyses. There are a few sentences that could be rewritten to clarify meaning and improve reader comprehension. My comments and questions are the following:

Comprehension or interpretation-based questions or comments:

1. The authors exclude case-control studies on the basis that samples included in case-control studies are non-representative. However, based on the sampling scheme of a case-control study, the controls represent the underlying exposure distribution in the population. Since many studies examining use of medications during pregnancy are case-control studies (particularly comparative studies of exposure to medications during pregnancy and rare outcomes such as congenital anomalies), are the authors concerned that they may be missing crucial information to inform prevalence estimates?

2. There were 18,418 titles and abstracts screened and of those, 164 were assessed for eligibility. The authors should describe the reason(s) fewer than 10% of the titles and abstracts screened were assessed for eligibility, especially since they made no exclusions for time, date or language in their search terms. For example, did the authors exclude other systematic reviews on the prevalence antibiotic use during pregnancy (e.g., Kuperman AA, Koren O. Antibiotic use during pregnancy: how bad is it? BMC Med. 2016;14(1):91. Published 2016 Jun 17. doi:10.1186/s12916-016-0636-0)?

3. The authors include a through line at ~28% prevalence in Figure 3. Is there a particular reason for this? It wasn’t clear to the reviewer when reading the methods section

4. The authors mention in the discussion section that “there is a chance of underestimation regarding primary data study estimates compared to secondary dataset studies, and these need to be properly interpreted. The last one is recorded prospectively and independently, avoiding maternal recall bias”. First, this sentence is difficult to understand. When stating the “last one” are the authors referring to the former (primary data) or the latter (secondary data)? Second, maternal recall bias is not simply a result of retrospective data collection, but rather a particular scenario were recall of the exposure is differential by the outcome. Thus, it is unclear how simply using secondary data would lead to recall bias in this case. It may lead to recall error but not bias.

5. The authors state that studies using both primary and secondary data sources have to deal with misclassification, since prescription and dispensing information may not reflect actual use. Again, the authors seem to be conflating terms that may lead to confusion for the reader. Since the authors present a descriptive analysis rather than a comparative one (with association or effect measures), it seems more likely that they are referring to measurement error rather than bias. If the authors do in fact believe that there would be bias (instead of simply error), they should be specific about the type of misclassification that would occur in these scenarios. For example, in a prospective study, they are more likely to incur non-differential misclassification of the exposure (i.e., use of antibacterials during pregnancy).

6. The authors state that differences in the range of prevalence estimates may be due to differences in calculating gestational age. While that may certainly play a role, the authors should also specifically address how the different exposure windows may affect prevalence estimates. Evaluating prevalence of use of antibacterials during the 1st, 2nd and 3rd trimester, is predicated on pregnancies continuing for longer gestational periods and thus have a greater opportunity to be exposed to antibiotics (compared than pregnancies that result in early losses or preterm delivery).

7. In the discussion section, the authors state the “there is a chance of under-detection bias by antibacterial over-the-counter (OTC) sales” when discussion a study using data from the Tennessee Medicaid Program. As the reviewer is aware, there are no systemic antibacterials that are approved or available for OTC dispensing in the US.

8. In the discussion section, the authors state “since the database may not cover a large segment of the population not covered, given the lack of data from the public sector”, again referring to data from Tennessee Medicaid Program. As the reviewer is aware, Medicaid is a joint federal and state program which gives health coverage to people with limited income and would be considered the public coverage option.

9. “as to comparative measures – relative risk or odds ratio - proportional data is intrinsically non-comparative, leading to more variability of estimates among different studies” This sentence is difficult to understand. While risk ratios and odd ratios are not interchangeable, they are comparable. And in some cases, an odds ratio may approximate a risk ratio. Therefore, it was unclear to the reviewer what the authors meant comparing proportional data to comparative measures in this sentence.

Minor/Editorial comments

1. In the introduction, the authors write that the inappropriate use of antibacterial can lead to antibacterial resistance (AMR). AMR refers to antimicrobial resistance (resistance to antimicrobials, includes resistance to antivirals, antifungals and antiparasitics, in addition to antibacterials while antibacterial resistance only refers to resistance to the latter). Please correct this in the manuscript

2. Please remove the article “The” before “The AMR” in the second sentence of the second paragraph in the introduction

3. Please consider removing the article numbers in the text of the results section when describing the proportion of articles that provided information on various characteristics (e.g., maternal age). For example, the sentence “regarding the 79 included studies, 43 provided information on maternal age [listed article numbers]” should have the listed article numbers removed. The authors may consider simply referencing table S2

6. PLOS authors have the option to publish the peer review history of their article (what does this mean?). If published, this will include your full peer review and any attached files.

Reviewer #1: No

Reviewer #2: No

---

## [Author Response · Author response to Decision Letter 0]

9 Feb 2024

On behalf of the authors, I would like to thank both reviewers and the academic editor for their considerations. Also, I would like to inform you that we have an institutional budget of 6,000 Brazilian reais to cover the cost of the publication. This budget is date-restrictive, with a deadline of March 31st, 2024. Otherwise, we have an institutional budget of 1,500 Brazilian reais to cover the cost of the publication without a deadline. Our university is not covered by the PLOS institutional partners. We would be grateful for the opportunity to publish our work in PLOS One. Best regards

---

## [Decision Letter · Decision Letter 1]

18 Jun 2024

PONE-D-23-25313R1Prevalence of systemic antibacterial use during pregnancy worldwide: a systematic reviewPLOS ONE

Dear Dr. Guimarães,

Thank you for submitting your manuscript to PLOS ONE. After careful consideration, we feel that it has merit but does not fully meet PLOS ONE’s publication criteria as it currently stands. Therefore, we invite you to submit a revised version of the manuscript that addresses the points raised during the review process.

We have received conflicting decisions from the reviewers regarding your manuscript. We suggest that you revise your manuscript and address all the comments provided by the reviewer, particularly those that have been claimed as not yet addressed.

We look forward to receiving your revised manuscript.

Kind regards,

Nur Aizati Athirah Daud, Ph.D.

Academic Editor

PLOS ONE

Reviewers' comments:

Reviewer's Responses to Questions

**Comments to the Author**

1. If the authors have adequately addressed your comments raised in a previous round of review and you feel that this manuscript is now acceptable for publication, you may indicate that here to bypass the “Comments to the Author” section, enter your conflict of interest statement in the “Confidential to Editor” section, and submit your "Accept" recommendation.

Reviewer #1: (No Response)

Reviewer #2: All comments have been addressed

2. Is the manuscript technically sound, and do the data support the conclusions?

Reviewer #1: Yes

Reviewer #2: Yes

3. Has the statistical analysis been performed appropriately and rigorously? 

Reviewer #1: N/A

Reviewer #2: N/A

4. Have the authors made all data underlying the findings in their manuscript fully available?

Reviewer #1: Yes

Reviewer #2: Yes

5. Is the manuscript presented in an intelligible fashion and written in standard English?

Reviewer #1: Yes

Reviewer #2: Yes

6. Review Comments to the Author

Reviewer #1: The revised version of the manuscript did not respond to my previous comments. In addition, there was no response to each of the comments.

In the current version, while the introduction provides valuable context and highlights important issues related to antibacterial use during pregnancy and antimicrobial resistance (AMR), it does not sufficiently justify the specific objectives of the study. It would benefit from a clearer linkage between the background information and the study's aims. Specifically, it should better articulate why it is important to perform a descriptive analysis of study-level characteristics. Additionally, it should explain how these findings will contribute to addressing the gaps in knowledge, particularly in low- and middle-income countries (LMICs) as the results also included non-LMICs

the resolution of Figure 1 to 4 are poor. very hard to read

Reviewer #2: (No Response)

7. PLOS authors have the option to publish the peer review history of their article (what does this mean?). If published, this will include your full peer review and any attached files.

Reviewer #1: No

Reviewer #2: No

---

## [Author Response · Author response to Decision Letter 1]

2 Aug 2024

Manuscript ID: PONE-D-23-25313R1

Prevalence of systemic antibacterial use during pregnancy worldwide: a systematic review

We appreciate the thorough evaluation of our work by the reviewers. The word "Response" precedes each response to the comments. We believe the modifications made based on these comments have substantially improved the manuscript. Furthermore, we are available to answer any additional questions, doubts, or clarifications you may have.

Reviewer #1: The revised version of the manuscript did not respond to my previous comments. In addition, there was no response to each of the comments.

Response: Dear reviewer, considering the commentary on the manuscript and the response letter from the first round of revision, we thoroughly examined the response letter to identify any unanswered questions. Additionally, we reviewed the manuscript's text to ensure the response letter aligns with the revised version.

Our answers may have been justifications for not following some recommendations, as in the case of the first presented comment: “Firstly, it would be beneficial to examine the prevalence of antibiotic use both before and after the implementation of the WHO's AMR global plan. Additionally, investigating the common indications for antibacterial usage and stratifying prevalence based on these indications could provide valuable context.”

In light of the second review, we have revised our response to this commentary. We included analyses to consider the first suggestions, which we initially decided not to perform due to the small number of studies in the resulting subgroups. In this second-reviewed version, we describe the prevalence of antibacterial use before and after the WHO's AMR global plan was implemented. We included the following text in the Results section: “Subsequently to the Global Action Plan on Antibacterial Resistance stated by WHO in 2016 [8], only three studies [28,52,85] were performed using data collection since 2016, and the proportions ranged from 2.6% (95%CI not reported) to 8.2% (95%CI not reported). Studies with data collected before 2016 showed a prevalence ranging from 2.0% (95%CI 2.0–2.0) to 64.3% (95%CI not reported).” Similarly, we identified only six studies with information about indications, and we reported the most common indications in the Results section, as follows: “Only six studies [11,23,25,28,47,81] provided information on antibacterial indications. Among those, the most common indications were urinary tract infections and respiratory tract infections.”. We reported both analyses in the Methods section: “We described proportions of antibacterial use during pregnancy before and after the Global Action Plan on Antibacterial Resistance stated by the WHO. We identified the most common indications for antibacterial use during pregnancy among the included studies.” Unfortunately, additional analyses were not feasible due to the small number of studies. Finally, we reviewed all comments and answers from the first round of revisions. Any further unclear points or concerns identified by the reviewer will be addressed. We would be pleased to revisit and make additional adjustments to the text as necessary.

Reviewer #1: In the current version, while the introduction provides valuable context and highlights important issues related to antibacterial use during pregnancy and antimicrobial resistance (AMR), it does not sufficiently justify the specific objectives of the study. It would benefit from a clearer linkage between the background information and the study's aims. Specifically, it should better articulate why it is important to perform a descriptive analysis of study-level characteristics. Additionally, it should explain how these findings will contribute to addressing the gaps in knowledge, particularly in low- and middle-income countries (LMICs) as the results also included non-LMICs

Response: Thank you for your comment. We underlined the importance of the descriptive analysis of study-level characteristics, including the following text in the introduction section: A tabular summary of prevalence point estimates can provide useful information of evidence synthesis [12]. Thus, gathering data on the proportion of antibacterial use during pregnancy enables the comparability between subgroups of studies, in order to identify point estimates variability according to relevant characteristics, such as study type, data source and country income”. Also, we added in the introduction section information about the relevance of the results focused on LMICs: “Despite this difference, estimates from LMICs can provide an overview of the likely prevalence of antibacterial use during pregnancy, addressing the needs of the AMR action plans in LMICs [6,8]. Further information on trends of antibacterial use and indication is also needed.”

Reviewer #1: The resolution of Figure 1 to 4 are poor. very hard to read

Response: As suggested, we improved the resolution of figures using the Preflight Analysis and Conversion Engine (PACE) digital diagnostic tool.

---

## [Editor Report · Decision Letter 2]

19 Aug 2024

Prevalence of systemic antibacterial use during pregnancy worldwide: a systematic review

PONE-D-23-25313R2

Dear Dr. Guimarães,

We’re pleased to inform you that your manuscript has been judged scientifically suitable for publication and will be formally accepted for publication once it meets all outstanding technical requirements.

Kind regards,

Nur Aizati Athirah Daud, Ph.D.

Academic Editor

PLOS ONE
---

## [Editor Report · Acceptance letter]

28 Aug 2024

PONE-D-23-25313R2 

PLOS ONE

Dear Dr. Guimarães, 

I'm pleased to inform you that your manuscript has been deemed suitable for publication in PLOS ONE. Congratulations! Your manuscript is now being handed over to our production team.

Kind regards, 

on behalf of

Dr. Nur Aizati Athirah Daud 

Academic Editor

PLOS ONE